# Underlying Conditions and Clinical Spectrum of Chronic Pulmonary Aspergillosis (CPA): An Experience from a Tertiary Care Hospital in Karachi, Pakistan

**DOI:** 10.3390/jof6020041

**Published:** 2020-03-26

**Authors:** Nousheen Iqbal, Muhammad Irfan, Ammar Mushtaq, Kauser Jabeen

**Affiliations:** 1Department of Medicine, Jinnah Medical and Dental College, Karachi 74800, Pakistan; naush.akuh@gmail.com (N.I.); ammarrajbhai@gmail.com (A.M.); 2Section of Pulmonary and Critical care Medicine, Department of Medicine, Aga Khan University, Karachi 75300, Pakistan; muhammad.irfan@aku.edu; 3Department of Pathology& Laboratory Medicine, Aga Khan University, Karachi 75300, Pakistan

**Keywords:** chronic pulmonary aspergillosis, structural lung diseases, developing country, Pakistan

## Abstract

The incidence of chronic pulmonary aspergillosis (CPA) is especially increasing in high tuberculosis (TB) burden countries. Despite a high estimated CPA burden in Pakistan, actual data on CPA are not available. The aim of the current study is to determine the underlying conditions and clinical spectrum of CPA at a tertiary care hospital in Karachi, Pakistan. This is a retrospective chart review study in patients admitted with CPA from January 2012 to December 2017. A total of 67 patients were identified during the study period. Mean age of CPA patients was 45.9 ± 15 years, 44 (65.7%) were male and 19 (28.4%) had diabetes. The most common type of CPA was simple aspergilloma (49.2%) followed by chronic cavitary pulmonary aspergillosis (CCPA) (44.7%). TB was the underlying cause of CPA in 58 (86.6%) patients followed by bronchiectasis caused by allergic bronchopulmonary aspergillosis (ABPA) 8 (11.9%). *Aspergillus flavus* was identified in 17 (47.2%), followed by *A. fumigatus* in 13 (36.1%) CPA patients. Isolation of multiple *Aspergillus* species was found in 10 (25.6%) patients. Itraconazole was given in 27 (40.3%) patients and a combination therapy of itraconazole and surgery was given in 21 (31.34%) patients. We found aspergilloma and CCPA as the most prevalent forms of CPA in our setting. Further large prospective studies using *Aspergillus* specific immunoglobulin G (IgG) antibodies testing are required for better understanding of CPA in Pakistan.

## 1. Introduction

Chronic pulmonary aspergillosis (CPA) is increasingly being recognized as an infectious post-tuberculosis (TB) sequelae [1,2]. The spectrum of CPA includes simple aspergilloma, chronic cavitary pulmonary aspergillosis (CCPA), aspergillus nodules, sub-acute invasive aspergillosis (SAIA), and chronic fibrosing pulmonary aspergillosis (CFPA) [3,4]. In addition to TB, CPA can also affect patients with other pulmonary disorders such as cystic fibrosis, bronchiectasis, sarcoidosis, chronic obstructive pulmonary diseases (COPD), pulmonary fibrosis, healed abscess cavities, and prior infection by non-tuberculous mycobacteria [4,5]. *Aspergillus fumigatus* is mainly reported as a causative species, but non-*A. fumigatus* species such as *A. niger* and *A. flavus* infection have also been reported from many countries [6,7]. It has been estimated that patients with residual pulmonary cavities of ≥2 cm after TB treatment have a 20% chance of developing aspergillomas [8]. Recent studies conducted in high TB burden settings such as India and Uganda evaluating CPA in post-TB patients estimated CPA to be more common in cavitary disease [9,10]. Another study from Cuba also showed an association between high levels of *Aspergillus* immunoglobulin G (IgG) and the presence of cavities [11]. If left untreated, CPA is associated with significant morbidity and mortality [3,4,5].

Although Pakistan has a huge burden of TB, CPA still exists as an under-recognized entity in Pakistan with limited diagnostic and treatment expertise. A recent burden estimate of CPA from Pakistan reported a high prevalence of 39 cases per 100,000 people [12]. It is difficult to diagnose CPA in Pakistan due to the non-availability of *Aspergillus*-specific IgG and limited knowledge of clinicians regarding this disease entity; therefore, CPA is often misdiagnosed. A study evaluating CPA burden in active TB patients in Indonesia reported that around 18% of CPA patients are misdiagnosed as TB [13]. Recently modified criteria have been suggested for low-resource/income countries which define CPA on the basis of presence of one or more cavities with or without a fungal ball or nodules/pleural thickening on thoracic imaging, direct evidence of *Aspergillus* infection (microscopy or culture) or an immunological response to *Aspergillus* spp. and exclusion of alternative diagnoses, all present for at least three months along with chronic respiratory symptoms [14].

Clinical data on underling conditions, clinical characteristics, and spectrum of CPA from Pakistan, a high TB-burden country, are not available. Therefore, this study aims to report the underlying conditions and types/spectrum of CPA from a tertiary care hospital in Karachi, Pakistan. This study provides some baseline knowledge that will help for better understanding and for future studies in Pakistan.

## 2. Material and Methods

### 2.1. Study Design and Settings

This was a retrospective chart review study conducted at Aga Khan University Hospital, Karachi, Pakistan from January 2012 to December 2017. All adult patients (>18 years) diagnosed as aspergillosis using the International Classification of Disease, Ninth Revision codes (ICD-9 1173) were identified from the medical records. CPA was diagnosed by using the diagnostic criteria proposed by Denning et al. for the resource constrained settings [14] as *Aspergillus*-specific IgG testing was not performed. CPA was further classified into (1) simple aspergilloma, (2) CCPA, (3) CFPA, (4) *Aspergillus* nodules, and (5) SAIA, based on definitions from Denning et al. [3]. Patients with invasive pulmonary and extra-pulmonary aspergillosis were excluded from the study. Patients who did not fulfill the above-defined criteria and had incomplete records were also not included. Data were collected on predesigned proforma with details of patient demographics, associated conditions, underlying lung pathologies, and radiological, microbiological, and treatment data. The ethical review committee of the Aga Khan University Hospital approved the study protocol before the commencement of the study. 

### 2.2. Statistical Analysis

All analyses were conducted by using the SPSS (Release 19.0, standard version, copyright © SPSS; 1989–2002). A descriptive analysis was performed for demographic data presented as mean ± SD, for quantitative variables like age. Numbers (percentage) were calculated for qualitative variables; that is, gender, mortality, smokers, associated underlying disease, radiographic findings, isolated *Aspergillus* species, and respiratory complications.

## 3. Results

A total of 350 records were reviewed with the diagnosis of aspergillosis and 67 (19.1%) patients (male, 44; female, 23) fulfilled the criteria of CPA (Figure 1). Mean age of CPA patients was 45.9 ± 15 years (minimum 22 years and maximum 78 years). Cough was the predominant symptom noticed in 55 (82.0%) CPA patients followed by hemoptysis in 40 (59.7%) and weight loss in 34 (50.7%). Clinical, radiographical, and microbiological criteria were positive in 33 (49.2%) patients. Diabetes (DM) was present in 19 (28.4%) patients as an associated condition. Amongst the underlying lung conditions, previous TB was present in 58 (86.6%) CPA patients followed by bronchiectasis secondary to allergic bronchopulmonary aspergillosis (ABPA) in 8 (11.9%) patients (Table 1). The most common form of CPA was simple aspergilloma (*n* = 33; 49.2%) followed by CCPA (*n* = 30; 44.7%). Culture was positive in 39 patients and *Aspergillus flavus* was the most frequently isolated species (*n* =17/39, 47.2%) followed by *A. fumigatus* (*n* = 13/39, 33.3%) and *A. niger* (*n* = 12/39, 30.7%). In 10 (25.6%) patients there was isolation of two or more *Aspergillus* species (Table 2). Itraconazole alone was used for treating 27 (40.3%) CPA patients and voriconazole alone in four (5.9%). Surgical excision with itraconazole therapy was administered in 21 (31.34%) patients and 9 (13.4%) patients underwent surgery alone. Five patients (7.4%) did not receive any treatment.

## 4. Discussion

This is the first study from Pakistan describing clinical spectrum and underlying associated conditions of CPA. We found that previous TB was the most common underlying lung condition and aspergilloma and CCPA were the most frequent types of CPA. A significant burden of CPA has been reported globally with most cases estimated in high TB burden countries. Actual data of CPA is not available from Pakistan, however recent estimates showed a burden of 39 cases/100,000 people [5,6]. Similar to the data from other countries, CPA in our cohort of patients also occurred most commonly in post-TB patients [3,4,15]. Data from India also suggested a significant burden of CPA in post-TB and asthmatic patients [2]. We also found post-ABPA bronchiectasis as an underlying cause in 11.9% of our CPA patients. Contrary to the countries where TB is endemic, COPD has been reported to be the most common risk factor of CPA in other countries [3,4,5]. Prior cardiothoracic surgery has also been reported to increase the risk of CPA [4,16,17]. In addition to TB, we also found COPD and previous cardiothoracic surgery as underlying conditions of CPA in our study population. 

Diabetes remains an important associated disease in TB patients and is related with aggressive course and outcomes [18,19,20]. We found diabetes in 28.4% of our patients with CPA. It is well established through literature that diabetes is associated with cavitation, treatment failure, and relapses in TB patients. Patients with DM are prone to develop *Aspergillus* infection because of impaired immunity and greater structural lung damage [20]. The most prevalent form of CPA in our study was aspergilloma followed by CCPA. Literature also suggest these as the most common manifestations of CPA which can progress to CFPA if they remain untreated [3].

*Aspergillus*-specific IgG is usually positive in >90% of cases with CPA [3,14]. It has good positive predictive value and can differentiate infected from colonized individuals [21]. Unfortunately, this test is not available currently in our country and therefore IgG levels were not tested in these patients. The most common *Aspergillus* species in our study was *A. flavus* followed by *A. fumigatus* and *A. niger*. Data suggest *A. fumigatus* is the most frequently isolated species seen in CPA patients with *A. niger* and *A. flavus* as rare causes [4,5]. However, in many settings non-*A. fumigatus* species are a predominant cause of CPA similar to our patient population and therefore concerns have been raised regarding false negative IgG results in such populations [6,7]. The majority of our patients received either itraconazole or a combination of surgery with itraconazole. Due to its low cost, itraconazole is usually the first choice of treatment for CPA in our country, followed by voriconazole with a response rate of more than 50% [22,23]. The majority (33; 49.2%) of our patients underwent a lobectomy for uncomplicated aspergilloma. 

This is the first report from Pakistan which studied the clinical characteristics of CPA patients. However, there were several limitations of this data. Firstly, this was a retrospective single center study. Secondly, due to non-availability, *Aspergillus* specific IgG was not detected, and the actual burden of CPA could be higher. Additionally, we were unable to see outcomes of these patients because of retrospective chart review. Time duration of developing CPA was not addressed in this study because of limitations of data availability. Finally, year-wise incidence could not be calculated as data regarding total number of TB patients or other baseline hospital parameters (such as patient days or 1000 discharges) were not available.

## 5. Conclusions

The majority of CPA patients are usually misdiagnosed as recurrence of TB due to similarities in clinical presentation. Additionally, the absence of an *Aspergillus* IgG test leads to further delay in diagnosis. Therefore, it is important to recognize CPA and raise awareness amongst clinicians in Pakistan where TB is endemic and CPA burden is been estimated to be high. Further large-scale prospective studies using *Aspergillus-*specific IgG testing are urgently required for better understanding of the prevalence of CPA in post-TB and other structural lung disease patients. 

## Figures and Tables

**Figure 1 jof-06-00041-f001:**
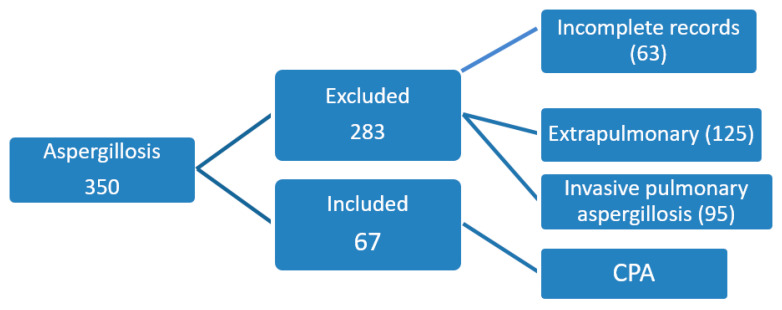
Patient enrollment.

**Table 1 jof-06-00041-t001:** Baseline characteristics of patients and diagnostic criteria of chronic pulmonary aspergillosis (CPA).

Variables	Total *n* = 67	Percentage
**Age (Mean)**	45.9 ± 15 years	
**Gender**		
Male	44	65.7%
Female	23	34.3%
**Smoking status**		
Ex-smoker	21	31.3%
Non-Smoker	46	68.7%
Current smoker	0	-
**Clinical Symptoms**		
Cough	55	82.0%
Hemoptysis	40	59.7%
Weight loss	34	50.7%
Dyspnea	24	35.8%
Fever	22	32.8%
Anorexia	7	10.4%
Chest pain	9	13.4%
**Associated Condition**	*n* = 33	49.3%
DM	19	28.4%
CLD	2	3.0%
CRF	2	3.0%
Malignancy	3	4.5%
Inhaled steroid	14	20.9%
Oral steroid	11	16.4%
**Underlying Lung Condition**		
Previous TB	58	86.6%
ABPA bronchiectasis	8	11.9%
Post-pneumonia bronchiectasis	6	8.9%
COPD	7	10.4%
Previous CTS surgery	5	7.4%
Active TB	2	2.9%
ILD	2	2.9%
Sarcoidosis	1	1.4%
Granulomatous polyangiitis	1	1.4%
**Diagnostic criteria**		
Clinical + Radiographic+ Histopathology	28	41.7%
Clinical + Radiographic + Microbiology	35	49.2%
Clinical + Radiographic+ Histopathology + Microbiology	6	8.9%

CTS: cardiothoracic surgery, DM: diabetes, CLD: chronic liver disease, CRF: chronic renal failure, TB: tuberculosis, ABPA: allergic bronchopulmonary aspergillosis, COPD: chronic obstructive pulmonary disease, ILD: interstitial lung disease.

**Table 2 jof-06-00041-t002:** Type and microbiological features for patients with CPA.

	*n* = Number	Percentage (%)
**Types of CPA**		
Simple aspergilloma	33	49.2%
CCPA	30	44.7%
CFPA	3	4.4%
SAIA	1	1.4%
***Aspergillus* species**	*n* =39	
*A. fumigatus* *A. flavus* *A. niger* *A. terreus*	13 17 12 7	33.3% 43.5% 30.7% 17.9%
**Two or more *Aspergillus* species** *A flavus + A. niger + A. terreus* *A. flavus + A. niger* *A. flavus + A. niger + A. fumigatus*	8/39 3 4 1	20.5% 20.0% 40.0% 10.0%

CCPA: chronic cavitary pulmonary aspergillosis, CFPA: chronic fibrosing pulmonary aspergillosis, SAIA: sub-acute invasive aspergillosis.

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
