# Peer review of "Underlying Conditions and Clinical Spectrum of Chronic Pulmonary Aspergillosis (CPA): An Experience from a Tertiary Care Hospital in Karachi, Pakistan"

_jof, 2020, doi:10.3390/jof6020041_

Round 1
Reviewer 1 Report
This is a retrospective study describing the clinical characteristics of 67 CPA cases in a university hospital in Pakistan from 2012 to 2019. This is a nice information to publish, but a couple of information could make it more valuable for the topic.
Here my comments:
- Authors highlighted the CPA occurring after TB. If they have the number of CPA cases per year from 2012 to 2019, is it possible for them to have the number of cases of TB and estimate incidence of CPA in TB patients per year? That would make possible to compare real-life data with the previously published estimates they mention in the manuscript.
- Also, I think it would be good to know the time between the TB diagnosis and the diagnosis of CPA.
- Finally, incidence of CPA by 1000 discharges or patients-day could be estimated if the authors have access to that information
Author Response
1. Authors highlighted the CPA occurring after TB. If they have the number of CPA cases per year from 2012 to 2019, is it possible for them to have the number of cases of TB and estimate incidence of CPA in TB patients per year? That would make possible to compare real-life data with the previously published estimates they mention in the manuscript.
Response
Unfortunately, we do not have access to this data and could not add this to the manuscrip
2. Also, I think it would be good to know the time between the TB diagnosis and the diagnosis of CPA.
Response
Unfortunately, we do not have access to this data and could not add this to the manuscript
3. Finally, incidence of CPA by 1000 discharges or patients-day could be estimated if the authors have access to that information
Response
Unfortunately, we do not have access to this data and could not add this to the manuscript
Reviewer 2 Report
This manuscript is a retrospective analysis of the estimated incidence of chronic pulmonary aspergillosis over a six year period at a tertiary hospital in Karachi, Pakistan. The authors postulate that "there is limited data available on CPA as a post-TB sequel and in structural lung disease", although I would argue that CPA following pulmonary TB and in structural lung disease are both well-known associations. Still, the Authors note that the burden of CPA in Pakistan is poorly-recognized due to poor-availability of the Aspergillus specific IgG so I think this manuscript is of interest in this setting.
A more thorough review of the literature would be beneficial, especially pertaining to the incidence/prevalence of CPA following TB infection in low resources settings such as Pakistan and in settings where the Aspergillus specific IgG is unavailable, as well as of other studies in these setting which have used the Denning criteria.
If accepted I would advise the authors to carefully proofread the manuscript as there are some spelling mistakes (line 37 - cavitory, line 19 - sequel, etc). It would also benefit from extensive English language editing and review of grammer.
It could be the PDF on my computer but it appears the text and boxes don't line up in Figure 1 as well as numbers and percentages in Table 1 for "Clinical Symptoms" (numerically there are 7 categories of clinical symptoms but only 6 associated percentages). If these errors are present on the submitted manuscript they should be fixed.
The abbreviation key for Table 1 should be in alphabetical order
It would be interesting if the Authors have any idea why A flavus was the most commonly isolated Aspergillus spp in this study, as they note that A fumigatus is the most common cause of CPA.
Author Response
This manuscript is a retrospective analysis of the estimated incidence of chronic pulmonary aspergillosis over a six-year period at a tertiary hospital in Karachi, Pakistan. The authors postulate that "there is limited data available on CPA as a post-TB sequel and in structural lung disease", although I would argue that CPA following pulmonary TB and in structural lung disease are both well-known associations. Still, the Authors note that the burden of CPA in Pakistan is poorly recognized due to poor-availability of the Aspergillus specific IgG so I think this manuscript is of interest in this setting.
- A more thorough review of the literature would be beneficial, especially pertaining to the incidence/prevalence of CPA following TB infection in low resources settings such as Pakistan and in settings where the Aspergillus specific IgG is unavailable, as well as of other studies in these setting which have used the Denning criteria.
Response
Introduction section has been modified to include data from India, Uganda, Indonesia and Cuba (line 46-61).
- If accepted I would advise the authors to carefully proofread the manuscript as there are some spelling mistakes (line 37 - cavitory, line 19 - sequel, etc). It would also benefit from extensive English language editing and review of grammer.
Response
Spelling mistakes and language revisions done extensively.
- It could be the PDF on my computer but it appears the text and boxes don't line up in Figure 1 as well as numbers and percentages in Table 1 for "Clinical Symptoms" (numerically there are 7 categories of clinical symptoms but only 6 associated percentages). If these errors are present on the submitted manuscript they should be fixed.
Response
All errors have been corrected (see Fig 1 and Table 1)
- The abbreviation key for Table 1 should be in alphabetical order
Response
Done (see Table 1)
- It would be interesting if the Authors have any idea why A flavus was the most commonly isolated Aspergillus spp in this study, as they note that A fumigatus is the most common cause of CPA.
Response
Although A. fumigatus has been reported to be more common in some setting, but there are several settings where non-A. fumigatus species are more common. References have been added in the paper (See reference 6 and 7).
Round 2
Reviewer 2 Report
I commend the Authors on the revisions and editions that they have made to the manuscript.